# Liquid Biopsy and Other Non-Invasive Diagnostic Measures in PCNSL

**DOI:** 10.3390/cancers13112665

**Published:** 2021-05-28

**Authors:** Alexander Baraniskin, Roland Schroers

**Affiliations:** 1Department of Medicine, Hematology and Oncology, Ruhr University Bochum, Universitätsklinikum Knappschaftskrankenhaus Bochum GmbH, 44892 Bochum, Germany; roland.schroers@rub.de; 2Department of Hematology, Oncology and Palliative Care, Evangelisches Krankenhaus Hamm, 59063 Hamm, Germany

**Keywords:** liquid biopsy, PCNSL, microRNA, MYD88, CSF

## Abstract

**Simple Summary:**

Primary central nervous system lymphoma (PCNSL) is an uncommon disease accounting for around 3% of primary CNS tumors. PCNSL exhibits aggressive clinical behavior and has an overall poor prognosis. The clinical presentation is variable, and there are no specific symptoms. Despite progress in radiographic neuroimaging, stereotactic brain biopsy remains obligatory for definitive diagnosis. Advanced standard diagnostics, including CSF cytology and flow cytometry, have limited sensitivity. Accordingly, there is an urgent need to improve the diagnostic tools for PCNSL, including novel non-invasive procedures. The aim of this review is to present and discuss modern methods that have the potential to contribute standard clinical diagnostics within the next few years.

**Abstract:**

Primary central nervous system lymphoma is a rare but highly aggressive form of non-Hodgkin lymphoma that remains confined to the CNS neuroaxis. The diagnosis of PCNSL requires a high level of suspicion as clinical presentation varies depending on the involved CNS areas. Neurological symptoms and MRI findings may mimic gliomas, demyelinating lesions, or infectious and granulomatous diseases. Almost all PCNSL patients undergo invasive surgical procedures for definite diagnosis. Stereotactic biopsy is still the gold standard in achieving a diagnostic accuracy of 73–97%. Both the potential procedural morbidity and mortality, as well as the time to definite histopathologic diagnosis resulting in delays of treatment initiation, have to be considered. On the contrary, minimally invasive procedures, such as MRI, CSF cytology, and flow cytometry, still have limited value due to inferior specificity and sensitivity. Hence, novel diagnostic approaches, including mutation analyses (MYD88) in circulating tumor DNA (ctDNA) and the determination of microRNAs (miR-21, miR-19b, and miR-92) as well as cytokine levels (IL10 and IL6) in blood, cerebrospinal fluid (CSF), and vitreous fluid (VRF), move into the focus of investigation to facilitate the diagnosis of PCNSL. In this review, we outline the most promising approaches that are currently under clinical consideration.

## 1. Introduction

Primary central nervous system lymphoma (PCNSL) is a rare disease accounting for around 4% of primary central nervous system (CNS) tumors in immunocompetent patients [1]. PCNSL comprises an uncommon subgroup of non-Hodgkin lymphomas that are limited to the brain, eyes, leptomeninges, and, in rare cases, the spinal cord. The definition of PCNSL includes the lack of any systemic disease manifestation at diagnosis, and it has to be separated from secondary spread of systemic lymphoma to the CNS (secondary CNS lymphoma, SCNSL). Thus, parallel lymphoma involvement both inside and outside the CNS is defined as SCNSL. About 95% of PCNSLs belong to the group of diffuse large B-cell lymphomas, whereas the remaining 5% consist of T-cell, marginal zone, Burkitt, and lymphoblastic lymphomas [2]. PCNSL characteristically develops between the ages of 50 and 70, although the disease may also occur at younger and older ages [3]. Risk factors for the occurrence of PCNSL include congenital and acquired immunodeficiencies (especially HIV and post-transplant conditions). Compared to classic diffuse large B-cell lymphoma (DLBCL), PCNSLs exhibit a more invasive growth pattern and have an inferior prognosis [1].

Unlike other primary brain tumors, PCNSL responds favorably to chemotherapy and radiation. However, survival is usually worse as compared to systemic non-CNS lymphomas. Moreover, the prognosis of some patients, such as the elderly or patients with relapsed or refractory diseases, remains particularly poor.

Various pathogenetic mechanisms in PCNSL have been described, including dysregulations in signaling pathways of NF-kB, JAK/STAT, Toll-like receptors, and B-cell receptors. Frequently, mutations in specific genes, including MYD88 (35–80%), PIM1 (69%), TBL1XR1 (24%), TRDM1 (24%), BTG2 (29%), and PRDM1 (24%), contribute to disease pathogenesis [4].

## 2. Clinical Presentation

The clinical presentation of PCNSL is diverse, and the disease course is commonly subacute, depending on the involved CNS location. Frequent symptoms of PCNSL include cognitive dysfunction, personality changes, psychomotor slowing, and disorientation. Raised intracranial pressure, headaches, and focal symptoms occur in approximately 50% of cases. Brainstem and cerebellar signs, cranial nerve dysfunction, and seizures are observed less frequently [5].

## 3. Diagnostic Procedures

Patients who develop neurologic symptoms and deficits should urgently undergo diagnostic brain imaging. Magnetic resonance imaging (MRI) is the imaging modality of choice. Here, PCNSLs usually appear as solitary lesions (60–70%) or, less often, as multifocal diseases [5]. Lymphoma lesions are mostly located periventricular, involving the white matter of the centrum semiovale, the corpus callosum, or the basal ganglia. PCNSLs are typically iso-hypointense on T1-weighted imaging and iso-hypointense to gray matter on T2-weighted imaging. Hypercellularity of the lymphomatous lesions results in strong homogeneous patterns in approximately 85% of PCNSLs. The lesions are typically surrounded by a moderate amount of edema. It must be pointed out that in common clinical situations, MRI findings and neurological symptoms may imitate high-grade gliomas, tumefactive demyelinating lesions as a rare form of multiple sclerosis, neurosarcoidosis, or infectious and different granulomatous diseases.

Despite substantial improvements in radiographic techniques, neuroimaging patterns are suggestive but not diagnostic of PCNSL. Hence, definitive diagnosis must still be accomplished by stereotactic brain biopsy, including histopathology and immunohistochemical staining.

Stereotactic biopsy remains challenging and traumatic, with a reported failure rate of up to 35% [6]. Furthermore, significant complications, including hematomas, seizures, and cerebral edema, as well as biopsy-related mortality, are reported in about 1% of procedures [7,8]. However, our own experience shows that the proportion of successful brain biopsies in CNS lymphoma is above 90% with negligible morbidity. In some instances, a biopsy cannot be accomplished due to the deep location or small size of the lesion. Notably, stereotactic brain biopsies and subsequent histopathologic analyses are time consuming and may contribute to prolonged time to diagnosis and treatment. This may be relevant considering that diagnostic delay is a major challenge in CNS lymphoma management [9].

Another potential obstacle in the diagnostic process of lymphoma is the use of corticosteroids. Frequently, steroids are administered to control symptoms before a definite pathological diagnosis has been accomplished. Indeed, corticosteroids prevent definite (histopathological) diagnosis due to rapid apoptosis of lymphoma cells [10].

## 4. Standard CSF Diagnostics

Analysis of CSF is an obligatory part of the standard diagnostic approach, unless lumbar puncture is contraindicated, e.g., due to abnormal intracranial pressure or impaired CSF kinetics due to cerebral herniation and space occupying lesions with mass effect in the large posterior fossa, Arnold-Chiari malformation, or a high risk of bleeding. Indeed, cytopathologic analysis of the CSF is considered the gold standard for the diagnosis of leptomeningeal malignant disease [11], although modern technologies, including cellular immunophenotyping by flow cytometry, molecular genetics, and proteochemical analyses, have been further developed to support the diagnosis of leptomeningeal lymphoma. Nevertheless, only in a subset of patients can the diagnosis of CNS lymphoma be made definitively based on cellular CSF assessments, such as cytopathology and immunophenotyping (Table 1).

Accordingly, there is an unmet need to further develop advanced diagnostic tests to facilitate the diagnostic process. Although many studies addressing non-invasive diagnostics of PCNSL have been conducted in recent years, none of the new methods are thus far established in everyday clinical practice.

In this review, we intend to discuss the most promising non-invasive assays in cerebrospinal fluid (CSF) and in peripheral blood for the diagnosis of PCNSL with a clinical focus.

## 5. Cytology and Flow Cytometry

Leptomeningeal seeding frequently occurs in PCNSL and has been demonstrated in as many as 80% of patients, as reported by Onda et al. in post-mortem histopathology [13]. Sensitivity and specificity of cytopathology based solely on Pappenheim-stained cytospin samples are unsatisfactory (Table 1). Hence, additional immunocytochemistry is recommended to improve the lymphoma detection rate.

Canovi et al. compared the diagnostic accuracy of flow cytometry and cytomorphology for leptomeningeal involvement by lymphoid malignancies in a study that included the results of 27 trials [14]. Unfortunately, a significant heterogeneity was demonstrated: identical results of parallel flow cytometry and cytomorphology were in ranges between 0.3% and 42.9%. Specimens with positive flow cytometry and negative cytomorphology were described in 89% of the studies, whereas samples with positive cytomorphology but negative flow cytometry were reported in 48% [14]. In our own study, we analyzed CSF specimens by both cytopathology with conventionally stained slides and multiparameter flow cytometry [12]. We demonstrated sensitivities of 13.3% for cytopathology and 23.3% for flow cytometry [12].

It should be emphasized that CSF cells are fragile and must be processed within a short period following a lumbar tap. Alternatively, CSF cells can be preserved in cell stabilizing reagents.

## 6. Free Light Chains

The potential diagnostic utilization of kappa and lambda free immunoglobulin light chain (FLC) concentrations and ratios in the CSF has been examined in two studies [15,16] (Table 2). Hildebrandt et al. reported FLC concentrations/ratios in the CSF as encouraging biomarkers for the diagnosis of leptomeningeal lymphoma [15]. In our own study, FLC concentrations and ratios were analyzed in 21 patients with PCNSL and SCNSL [16]. Regardless of leptomeningeal lymphoma confirmed by cytopathology, distinctly increased FLC ratios were detected in 52% (11/21) of patients with CNS lymphomas as compared to control individuals suffering from different neurological diseases. Remarkably, increased FLC ratios in CSF were preferentially detected in patients with subependymal lymphoma spread as demonstrated by MRI [16] (Table 2).

## 7. IL-10 and IL-6

Interleukin-10 (IL-10) and its receptors are overexpressed in PCNSL and function as strong immunosuppressive and anti-inflammatory cytokines. They play different roles in the development and progression of lymphoma and inhibit apoptosis [30]. On the contrary, interleukin-6 (IL-6) is a pluripotent inflammatory factor that promotes lymphoid cell growth and regulates immune function [31].

Sasayama et al. reported that CSF concentrations of IL-10 and IL-6 are significantly increased in PCNSLs as compared to other brain tumors [17]. Setting a cut-off level of 9.5 pg/mL, the sensitivity and specificity for IL-10 were 71% and 100%, respectively [17]. Nguyen-Them et al. demonstrated that IL-10 CSF levels with a 4 pg/mL cut-off permitted differentiation of PCNSL from other neurologic diseases with a sensitivity of 88.6% and specificity of 88.9% [18]. In another study, Geng et al. reported a sensitivity of 59.4% for IL-10 at a 2.07 pg/mL cut-off [32]. Shao et al. defined an IL-10 cut-off level of 8.3 pg/mL, providing a sensitivity and specificity for the diagnosis of PCNSL of 59.0% and 98%, respectively. Interestingly, the inclusion of IL-6 in a combined ratio IL-10/IL-6 ratio at a cut-off of 1.6 increased the sensitivity and specificity to 66.0% and 91%, respectively [20]. Similar data were reported by Sasagawa et al. for IL-10 with a cut-off of 3 pg/mL, providing a sensitivity and specificity of 94.7% and 100%, respectively [19].

On the other hand, Ungureanu et al. reported lower CSF levels of IL-6 in PCNSL patients compared with CNS inflammatory lesions. The levels of IL-10 were significantly higher in the same group of PCNSL patients [33].

The most striking data were reported by Song et al.: CSF samples collected from 22 PCNSL patients were analyzed, and IL-10 at a cut-off of 8.2 pg/mL resulted in a sensitivity and specificity of 95.5% and 96.1%, respectively. Applying a CSF IL-10/IL-6 ratio with a cut-off value of 0.72 increased the sensitivity to 95.5% and the specificity to 100.0% [22].

In addition, Costopoulos et al. were able to differentiate patients with B-cell lymphoproliferative disorders from DLBCL measuring IL-10 and IL-10/IL6 ratios in CSF [23].

Even if the results of single studies appear promising, it must be noted that the results are hardly comparable due to various cut-off values (Table 2). Due to these limitations, the measurements of IL-10 and IL-10/IL6 currently cannot be recommended in routine clinical practice.

## 8. CXCL13

CXCL13 is a chemokine that plays an essential role in the homing of B-cells [34]. Rubenstein et al. analyzed CXCL13 concentrations in CSF samples of CNS lymphoma patients and reported a sensitivity of 69.9% and a specificity of 92.7% for levels above 90 pg/mL [35]. The combination of increased CXCL13 (>116 pg/mL) and increased IL-10 (>23 pg/mL) further improved the diagnostic value of both markers [35].

In another study, the combination of CSF CXCL-13 and CSF IL-10 together with impaired diffusion coefficients in cranial MRIs was examined. The study population comprised 38 PCNSL and 5 SCNSL patients. In summary, the authors described a sensitivity of 76.7% and a specificity of 90.9% for CXCL-13 at a cut-off value of >103 pg/mL, as compared to 44 patients in the control group suffering from different brain tumors or tumefactive demyelinating lesions [36].

It is undisputed that further studies with uniform cut-off values are necessary to pave the way for clinical usage of CXCL-13 (Table 2).

## 9. MYD88

The term “liquid biopsy” refers to the analysis of tumor-derived DNA from blood specimens to avoid an invasive tissue biopsy. Cell-free DNA fragments (cfDNA) are shed into the bloodstream from cells undergoing apoptosis and necrosis or are transferred into the blood via direct secretion of exosomes. CfDNA circulates in plasma at a low concentration.

To date, numerous molecular markers and genetic variants have been identified using peripheral blood samples in patients with primary brain tumors. The mutant allele frequency (MAF) should exceed 5% to reliably enable the detection of markers. Thus, the blood–brain barrier may hinder the detection of the required amount of cfDNA in plasma [37]. However, analysis of cfDNA in CSF could yield useful results, as cfDNA is almost exclusively of tumoral origin. Liquid biopsy offers the advantage of overcoming limitations encountered in the analysis of tissue biopsies, such as underrepresentation of the whole spectrum of genetic variants of a genetically heterogeneous tumor. In liquid biopsy, the “circulating” DNA is derived from multiple parts of the tumor and has the potential to represent the comprehensive mutational status of the whole tumor. Clearly, this only holds true under the assumption that all tumor areas secrete the same amount of DNA.

Concentrations of cfDNA vary between different lymphoma subtypes. The levels of cfDNA are higher in aggressive lymphomas compared to indolent lymphomas. Furthermore, tumor burden also affects cfDNA concentrations, which are consequently higher in advanced-stage diseases than in limited-stage diseases [38].

Recently, nucleic acid analysis by next-generation sequencing (NGS) has unraveled several genomic alterations in PCNSL [39]. Alterations in the nuclear factor-kappa B (NF-κB) pathway genes, MYD88, and CD79B have been identified in 40% to 80% of all cases [39,40].

MYD88 mutations are detectable in 58% to 86% of PCNSL cases [25,39,40]. Thus, MYD88 mutations can be regarded as a molecular marker for PCNSL. Specifically, the p.L265P amino acid substitution is the most frequent MYD88 mutation in CNS lymphoma. A single base substitution at c.794T > C results in an amino acid change from leucine to proline (L265P)) [41]. MYD88 acts as an adaptor protein transmitting signals from Toll-like receptors (TLRs) and activates IL-1 and IL-18 receptor signaling pathways. The MYD88 c.794T > C mutation also occurs in almost all cases of Waldenstrom macroglobulinemia and is rarer in systemic lymphomas [42]. Remarkably, the MYD88 c.794T > C mutation never occurs in tissue biopsies from non-hematologic brain tumors, such as glioblastoma or in solid metastatic tumors, suggesting that this mutation is a sensitive and specific biomarker for the differentiation of PCSNL from other CNS cancers [41]. The diagnosis of Waldenstrom macroglobulinemia can be ruled out by a bone marrow biopsy.

Interestingly, the MYD88 L265P variant has been reported to be associated with poor prognosis, especially in elderly patients [43].

Furthermore, detection of MYD88 L265P in cellular DNA from vitreous aspirates has been reported to improve the diagnosis of PCNSL. Bonzheim et al. have described the detection of MYD88 L265P in the majority (69%) of clinically, histologically, and molecularly confirmed vitreoretinal lymphoma. In addition, MYD88 L265P displays 100% specificity for the diagnosis of vitreoretinal lymphoma [21]. Therefore, the detection of MYD88 L265P allows a definitive lymphoma diagnosis even in poor quality samples. This represents a significant diagnostic advance in this challenging and rare entity.

Overall, the Achilles heel for the detection of specific mutations in blood, CSF, and the vitreous is low sensitivity (Table 2). The true positive rate is essentially dependent on two main parameters: the concentration of cfDNA and the method used for detecting mutations.

Currently, reverse transcription quantitative PCR (RT-qPCR) and panel next generation sequencing (NGS) are the techniques most often applied for the detection of MYD88 mutations [44], although samples containing a concentration of tumor DNA below the threshold required for either technique (NGS, 2–5%; RT-qPCR, 0.5%) may be rated as false negative [26]. In this context, droplet digital PCR (ddPCR) is a technique with a superior sensitivity for trace mutation identification compared to conventional PCR techniques [26,45,46].

Thus far, clinical or histological parameters in PCNSL patients that impact cfDNA shedding in plasma or CSF still remain unexplained. Overall, tumor volume has been demonstrated to contribute to the identification of tumor-derived alterations in blood cfDNA for nodal DLBCL [47]. Higher concentrations of cfDNA in the peripheral blood of patients with glioma are associated with an augmented contrast on MRI scans [28]. Another explanatory approach for the distinct false negative rate might be the application of corticosteroids subsequent to neurosurgical procedures and prior to peripheral blood collection. Relevant factors that impact the plasma shedding and measurement of cfDNA still remain unclear and should be investigated in further studies.

An important question in liquid biopsy of CNS tumors is whether plasma or CSF sampling is better for the detection of specific mutations in cfDNA of PCNSL. De Mattos-Arruda et al. demonstrated that cfDNA derived from the CNS was present at higher levels in CSF as compared to plasma in patients with malignant tumors limited to the CNS (glioblastoma, medulloblastoma, and CNS metastases of solid cancers). A possible explanation is that the cerebrospinal fluid is in intimate contact with tumor cells in CNS malignancies. Moreover, despite the disturbed blood–brain barrier, the proportion of cfDNA in the blood is too low to be detected.

## 10. IgH Gene Rearrangement

Testing for IgH gene rearrangements by PCR is an interesting diagnostic method of monoclonal B-lymphoma cell detection in CNS lymphoma. Sensitivities between 11% and 54% have been reported [29]. This PCR method may be particularly useful for small sample volumes and low CSF cell counts, in which flow cytometry is inconclusive.

## 11. microRNAs

microRNAs (miRNAs) are relatively small (18–24 nucleotides in length) non-coding RNAs that suppress gene expression post-transcriptionally by binding to target mRNA, triggering their degradation or translational downregulation. MiRNAs are involved in various biological processes, including cell proliferation, differentiation, metabolism, apoptosis, and tumorigenesis. Deregulated expression of miRNAs is commonly observed in human cancers. MiRNAs can also act as oncogenes or tumor suppressors. Therefore, expressions of individual miRNAs and miRNA signatures represent potent diagnostic and prognostic tools. Several miRNAs (miR-125a, miR-125b, miR-17-92 cluster, and miR-155) were illustrated to have a significant impact in the pathogenesis of DLBCL [29]. It has been shown, inter alia, that the miR-17-92 cluster, including miR19b and miR-92a, influences the expression of the tumor suppressors genes PTEN and of oncogene c-Myc [24,48].

In our own study, miRNA expression levels were determined in CSF samples collected from patients with PCNSL as compared to other neurologic disorders using TaqMan quantitative real-time PCR assays. Among six previously described candidate miRNAs (miR-15b, miR-19b, miR-21, miR-92a, miR-106b, and miR-204), the concentrations of miR-21, miR-19b, and miR-92a were significantly increased in the CSF of PCNSL patients compared with those from patients with other neurologic disorders. Accordingly, a diagnostic tree was suggested, starting with testing for miR-21 elevation, followed by testing for miR-19 and miR-92a increases. The diagnostic tree of these three CSF miRNAs represented a reliable diagnostic biomarker for PCNSL, with a diagnostic accuracy including 95.7% sensitivity and 96.7% specificity [49]. These data indicated that CSF miRNAs have the potential to be used as non-invasive diagnostic biomarkers for PCNSL. Notably, these findings could be reproduced (sensitivity 97.4%) in an enlarged cohort (*n* = 39) of PCNSL patients [50]. Furthermore, these data were confirmed by Zajdel et al. with a lower diagnostic accuracy, which can be explained by the selection of a different reference group [27].

Moreover, CSF concentrations of the above-mentioned miRNAs were significantly correlated with PCNSL disease status during therapy and/or disease follow-up, demonstrating their capability as biomarkers for treatment monitoring and follow-up [50]. In a meta-analysis, Wei et al. reported that miRNAs are precise and sensitive biomarkers for the diagnosis of PCNSL [51].

The role of microRNA determination in peripheral blood for the diagnosis of CNS lymphoma remains questionable. In our own studies, insignificant differences in serum microRNAs between PCNSL patients and control patients were observed [50]. On the contrary, Mao et al. demonstrated that miR-21 concentrations in serum were significantly higher in PCNSL as comparted to serum of control patients in two independent cohorts, with an AUC of 0.930 for the test cohort and 0.916 for the validation cohort. Furthermore, elevated serum concentrations of miR-21 could discriminate PCNSL from common brain tumors, including glioblastoma [52]. A correlation between miR-21 in serum and CSF was also illustrated in this study [52]. The difference between our data and the findings of Mao et al. may be caused by the varying composition of the respective control groups.

Previously, we analyzed the combination of CSF marker miR-21 with small nuclear RNA fragments of RNU2-1f in CSF [53]. For differentiation of PCNSL patients from controls with other neurologic disorders, we calculated a specificity of 95.7% and a sensitivity of 91.7% (AUC of 0.987) of this diagnostic approach.

Another study reported increased serum concentrations of miR-222 as a potential biomarker for early diagnosis of DLBCL and PCNSL in HIV-infected individuals [54].

It is possible that combinations of CSF- and serum-based miRNAs with currently available and promising biomarkers, such as MYD88 and CSF interleukin-10, could further improve diagnostic sensitivity and accuracy [55].

## 12. Conclusions

Combined “classical” CSF analysis, consisting of cytopathological analyses and immunophenotyping by flow cytometry, improves the diagnostic accuracy for PCNSL. However, the role of CSF markers indicating B-cell proliferation, including MYD88 mutational and microRNAs profiles, is increasing. These new biomarkers have the potential to improve the differential diagnosis of PCNSL. Nevertheless, validation and standardization as a prerequisite for routine clinical application are still pending.

## Figures and Tables

**Table 1 cancers-13-02665-t001:** Overview of standard diagnostic methods.

Biomarker/Method	Body Fluid	Number of Patients	Sensitivity (%)	Specificity (%)	References
cytology	CSF	37	13.30	-	Schroers et al. [12]
flow cytometry	CSF	37	23.3	-	Schroers et al. [12]

**Table 2 cancers-13-02665-t002:** Overview of the novel diagnostic approaches.

Biomarker/Method.	Number of Patients	Body Fluid	Sensitivity (%)	Specificity (%)	References
FLC concentrations/ratios	21	CSF	52.3	-	Schroers et al. [16]
**IL-10 (cut-off 9.5 pg/mL)**	66	CSF	71	100	Sasayama et al. [17]
**IL-10 (cut-off 4 pg/mL)**	119	CSF	88.6	88.9	Nguyen-Them et al. [18]
**IL-10 (cut-off 8.2 pg/mL)**	102	CSF	95.5	96.1	Song et al. [19]
**IL-10 (cut-off 8.3 pg/mL)**	108	CSF	59	98	Shao et al. [20]
**IL-10/IL-6 ratio (cut-off 1.6 pg/mL)**	108	CSF	66	91	Shao et al. [20]
**IL-10/IL-6 ratio (cut-off 0,72 pg/mL)**	102	CSF	95.5	100	Song et al. [19]
IgH gene rearrangement	32	CSF	54	97	Eckstein et al. [21]
CXCL13	220	CSF	69.9	92.7	Rubenstein et al. [22]
**Combination of CXCL13 and IL-10**	77	CSF	76.7	90.9	Mabray et al. [23]
MYD88	225	CSF	72	-	Ferreri et al. [24]
**Combination of MYD88 and IL-10**	225	CSF	94	98	Ferreri et al. [24]
MYD88	90	vitreous	69	100	Bonzheim et al. [25]
microRNA (miR-21, -19b, and -92)	53	CSF	95.7	96.7	Baraniskin et al. [26]
microRNA (miR-21, -19b, and -92)	53	CSF	63.3	80.7	Zajdel et al. [27]
**Combination of RNU2-1f and miR-21**	119	CSF	91.7	95.7	Baraniskin et al. [28]
miR-222	150	serum	80	82	Thapa et al. [29]

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
