# Peer review of "Liquid Biopsy and Other Non-Invasive Diagnostic Measures in PCNSL"

_cancers, 2021, doi:10.3390/cancers13112665_

Round 1

Reviewer 1 Report

Interesting review in the field of non surgical diagnosis of primary central nervous system lymphoma (PCNSL).

All the relevant strategies are listed and described. However, all the explanations seems to be listed in order to demonstrate that the best  approach is" miRNA detection" ... performed by the authors. A more balance demonstration should be done.

Three minor comments:

1-concerning the tables: number of studied cases in the different studies should be added.  These data should be useful since the number of cases were particularly low in the majority of the papers.

2- Other references could be added in order to proportionally increase the part dedicated to strategies other than miRNAs:

-Costopoulos (Sem Hematol 2018) demonstrate the role of the combined IL6 and IL10 in low grade NHL

-Ungureanu ((J Neurol 2021) demonstrate de role of IL6 to discriminate inflamatory pathologies of the CNS.

-Rimelen  (Acta Neuropathol commun 2019) describe a series of detection of MYD88 (L265P) in CNS using ddPCR in cellular and cf DNA fractions

3- The physiopathological reasons of these miRNAs expression could be added. What is their role in B lymphomagenesis? or in B cell differenciation? 

Author Response

We would like to thank the reviewers for the valuable comments on our manuscript. We addressed all criticized points and hope that the revised manuscript is now suitable for publication in Cancers. The modifications according to reviewer suggestions are marked in yellow and English revisions by a native English-speaking colleague are seen in the tracking modus.

Response to Reviewer #1:

  1. concerning the tables: number of studied cases in the different studies should be added.

Specific comments: The tables 1 and 2 were modified accordingly.

  1. Other references could be added in order to proportionally increase the part dedicated to strategies other than miRNAs

Specific comments: Thank you for the valuable hints. All mentioned references were included in the manuscript.

  1. The physiopathological reasons of these miRNAs expression could be added. What is their role in B lymphomagenesis? or in B cell differentiation?

Specific comments: We added the explanation of the relationship between the microRNAs and expression of oncogenes and tumor suppressor genes in DLBCL (page 8, lines 286-288).

Reviewer 2 Report

The submitted review:

Liquid Biopsy and Other Non-invasive Diagnostic Measures in PCNSL

covers the most important developments in the field of non-invasive approaches for differential diagnosis of PCNSL.

                I think that the Authors should correct or supplement the following information:

1/ lines: 297-299.

 The Authors wrote:

 “In a meta-analysis, Wei et al. reported, that compared to other CSF-based biomarkers, miRNAs were more precise and sensitive for the diagnosis PCNSL [42].”

In fact, the cited meta-analysis focuses on the diagnostic value of microRNAs for central nervous system cancers, without any comparison with other biomarkers.

2/ lines: 303-310.

The Authors discussed here in detail data from the study of Mao et al. in the context of their own unpublished data, but the cited study of Mao et al. has not been included in Table 2.

3/ lines: 245-247.

The Authors wrote:

In this context, droplet digital PCR (ddPCR) is a technique with a superior sensitivity for trace mutation identification compared to conventional PCR techniques [36].

The cited review “focus on the potential application of ddPCR and optimized NGS to detect ctDNA for detection of cancer recurrence and minimal residual disease as well as early diagnosis of cancer patients.”

The two following papers:

a/ Hiemcke-Jiwa LS, Minnema MC, Radersma-van Loon JH, Jiwa NM, de Boer M, Leguit RJ, de Weger RA, Huibers MMH. The use of droplet digital PCR in liquid biopsies: A highly sensitive technique for MYD88 p.(L265P) detection in cerebrospinal fluid. Hematol Oncol. 2018 Apr;36(2):429-435. doi: 10.1002/hon.2489. Epub 2017 Dec 6. PMID: 29210102.

b/ Hattori K, Sakata-Yanagimoto M, Suehara Y, Yokoyama Y, Kato T, Kurita N, Nishikii H, Obara N, Takano S, Ishikawa E, Matsumura A, Hasegawa Y, Chiba S. Clinical significance of disease-specific MYD88 mutations in circulating DNA in primary central nervous system lymphoma. Cancer Sci. 2018 Jan;109(1):225-230. doi: 10.1111/cas.13450. Epub 2017 Dec 23. PMID: 29151258; PMCID: PMC5765295.

are much more relevant, since directly concentrate on the use of ddPCR for detection of MYD88 mutation in liquid biopsies from PCNSL patients.

4. line 150 (Table 2) and lines: 289-294.

The Authors wrote:

“The diagnostic tree of these three CSF miRNAs represented a reliable diagnostic biomarker for PCNSL, with a diagnostic accuracy including 95.7% sensitivity and 96.7% specificity, respectively [40]. These data pointed out that CSF miRNAs have the potential to be used as non-invasive diagnostic biomarkers for PCNSL. Notably, in an enlarged cohort (n = 39) of PCNSL patients these findings could be confirmed (sensitivity 97.4%) [41].”

The Authors omitted here study by Zajdel et al.:

Zajdel M, Rymkiewicz G, Sromek M, Cieslikowska M, Swoboda P, Kulinczak M, Goryca K, Bystydzienski Z, Blachnio K, Ostrowska B, Borysiuk A, Druzd-Sitek A, Walewski J, Chechlinska M, Siwicki JK. Tumor and Cerebrospinal Fluid microRNAs in Primary Central Nervous System Lymphomas. Cancers (Basel). 2019 Oct 25;11(11):1647. doi: 10.3390/cancers11111647. PMID: 31731456; PMCID: PMC6895823.

where the utility of miR-21, miR-19b, and miR-92a as cerebrospinal fluid PCNSL markers has been confirmed, however with much lower accuracy (specificity of 80.77% and a sensitivity of 63.33%).

5. line 150 (Table 2)

 The Authors omitted also in their review two important studies:

Sasagawa Y, Akai T, Tachibana O, Iizuka H. Diagnostic value of interleukin-10 in cerebrospinal fluid for diffuse large B-cell lymphoma of the central nervous system. J Neurooncol. 2015 Jan;121(1):177-83. doi: 10.1007/s11060-014-1622-z. Epub 2014 Sep 26. PMID: 25258254.

Maeyama M, Sasayama T, Tanaka K, Nakamizo S, Tanaka H, Nishihara M, Fujita Y, Sekiguchi K, Kohta M, Mizukawa K, Hirose T, Itoh T, Kohmura E. Multi-marker algorithms based on CXCL13, IL-10, sIL-2 receptor, and β2-microglobulin in cerebrospinal fluid to diagnose CNS lymphoma. Cancer Med. 2020 Jun;9(12):4114-4125. doi: 10.1002/cam4.3048. Epub 2020 Apr 20. PMID: 32314548; PMCID: PMC7300423.

6. A comprehensive  systematic review on the diagnostic markers for CNS lymphoma in blood and cerebrospinal fluid has been published in 2018:

van Westrhenen A, Smidt LCA, Seute T, Nierkens S, Stork ACJ, Minnema MC, Snijders TJ. Diagnostic markers for CNS lymphoma in blood and cerebrospinal fluid: a systematic review. Br J Haematol. 2018 Aug;182(3):384-403. doi: 10.1111/bjh.15410. Epub 2018 May 29. PMID: 29808930; PMCID: PMC6099264.

I think that the Authors should mention the above paper and should also list the liquid biopsy markers for the diagnosis of central nervous system lymphoma described in reports published after 2018.

Author Response

We would like to thank the reviewers for the valuable comments on our manuscript. We addressed all criticized points and hope that the revised manuscript is now suitable for publication in Cancers. The modifications according to reviewer suggestions are marked in yellow and English revisions by a native English-speaking colleague are seen in the tracking modus.

Response to Reviewer #2:

  1. “In a meta-analysis, Wei et al. reported, that compared to other CSF-based biomarkers, miRNAs were more precise and sensitive for the diagnosis PCNSL [42].”

In fact, the cited meta-analysis focuses on the diagnostic value of microRNAs for central nervous system cancers, without any comparison with other biomarkers.

Specific comments: We thank the reviewer for the advice. We corrected the text accordingly (page 8; lines 305-306).

  1. The Authors discussed here in detail data from the study of Mao et al. in the context of their own unpublished data, but the cited study of Mao et al. has not been included in Table 2.

Specific comments:

  • Our mentioned data are published. The corresponding citation is included (page 8; line 310).
  • We did not include the data of Mao et al. in the table 2, because in this manuscript the data of sensitivity and specificity were not shown. Only ROC analyses were included.
  1. The Authors wrote:“In this context, droplet digital PCR (ddPCR) is a technique with a superior sensitivity for trace mutation identification compared to conventional PCR techniques [36].

The cited review “focus on the potential application of ddPCR and optimized NGS to detect ctDNA for detection of cancer recurrence and minimal residual disease as well as early diagnosis of cancer patients.”

The two following papers:

a/ Hiemcke-Jiwa LS, Minnema MC, Radersma-van Loon JH, Jiwa NM, de Boer M, Leguit RJ, de Weger RA, Huibers MMH. The use of droplet digital PCR in liquid biopsies: A highly sensitive technique for MYD88 p.(L265P) detection in cerebrospinal fluid. Hematol Oncol. 2018 Apr;36(2):429-435. doi: 10.1002/hon.2489. Epub 2017 Dec 6. PMID: 29210102.

b/ Hattori K, Sakata-Yanagimoto M, Suehara Y, Yokoyama Y, Kato T, Kurita N, Nishikii H, Obara N, Takano S, Ishikawa E, Matsumura A, Hasegawa Y, Chiba S. Clinical significance of disease-specific MYD88 mutations in circulating DNA in primary central nervous system lymphoma. Cancer Sci. 2018 Jan;109(1):225-230. doi: 10.1111/cas.13450. Epub 2017 Dec 23. PMID: 29151258; PMCID: PMC5765295.

are much more relevant, since directly concentrate on the use of ddPCR for detection of MYD88 mutation in liquid biopsies from PCNSL patients.

Specific comments: We have included both suggested papers in our revised manuscript.

  1. The Authors wrote: “The diagnostic tree of these three CSF miRNAs represented a reliable diagnostic biomarker for PCNSL, with a diagnostic accuracy including 95.7% sensitivity and 96.7% specificity, respectively [40]. These data pointed out that CSF miRNAs have the potential to be used as non-invasive diagnostic biomarkers for PCNSL. Notably, in an enlarged cohort (n = 39) of PCNSL patients these findings could be confirmed (sensitivity 97.4%) [41].”

The Authors omitted here study by Zajdel et al.: Zajdel M, Rymkiewicz G, Sromek M, Cieslikowska M, Swoboda P, Kulinczak M, Goryca K, Bystydzienski Z, Blachnio K, Ostrowska B, Borysiuk A, Druzd-Sitek A, Walewski J, Chechlinska M, Siwicki JK. Tumor and Cerebrospinal Fluid microRNAs in Primary Central Nervous System Lymphomas. Cancers (Basel). 2019 Oct 25;11(11):1647. doi: 10.3390/cancers11111647. PMID: 31731456; PMCID: PMC6895823.

where the utility of miR-21, miR-19b, and miR-92a as cerebrospinal fluid PCNSL markers has been confirmed, however with much lower accuracy (specificity of 80.77% and a sensitivity of 63.33%).

Specific comments:  We cited and discussed the mentioned paper. The key data of the paper were inserted in table 2.

  1. The Authors omitted also in their review two important studies: Sasagawa Y, Akai T, Tachibana O, Iizuka H. Diagnostic value of interleukin-10 in cerebrospinal fluid for diffuse large B-cell lymphoma of the central nervous system. J Neurooncol. 2015 Jan;121(1):177-83. doi: 10.1007/s11060-014-1622-z. Epub 2014 Sep 26. PMID: 25258254.

Maeyama M, Sasayama T, Tanaka K, Nakamizo S, Tanaka H, Nishihara M, Fujita Y, Sekiguchi K, Kohta M, Mizukawa K, Hirose T, Itoh T, Kohmura E. Multi-marker algorithms based on CXCL13, IL-10, sIL-2 receptor, and β2-microglobulin in cerebrospinal fluid to diagnose CNS lymphoma. Cancer Med. 2020 Jun;9(12):4114-4125. doi: 10.1002/cam4.3048. Epub 2020 Apr 20. PMID: 32314548; PMCID: PMC7300423.

Specific comments: The study of Sasagawa et al. was discussed and cited. The multi-marker algorithms by Maeyama et. was not mentioned, because it seems to be too complex to be used for clinical issues.

  1. A comprehensive systematic review on the diagnostic markers for CNS lymphoma in blood and cerebrospinal fluid has been published in 2018:

van Westrhenen A, Smidt LCA, Seute T, Nierkens S, Stork ACJ, Minnema MC, Snijders TJ. Diagnostic markers for CNS lymphoma in blood and cerebrospinal fluid: a systematic review. Br J Haematol. 2018 Aug;182(3):384-403. doi: 10.1111/bjh.15410. Epub 2018 May 29. PMID: 29808930; PMCID: PMC6099264.

I think that the Authors should mention the above paper and should also list the liquid biopsy markers for the diagnosis of central nervous system lymphoma described in reports published after 2018.

Specific comments: We respect the opinion of the reviewer; however, we see no convincing reason to cite the mentioned study. In our opinion, we have described the most promising and significant biomarker for PCNSL in CSF. We regard the “liquid biopsy markers for the diagnosis of central nervous system lymphoma described in reports published after 2018” not significant enough to be mentioned in our review.

Finally, the manuscript text has been revised and corrected by an English speaking native.